

# Oral history as a citizen science tool to understand biodiversity loss and environmental changes: on firefly extirpation in Morelia, Michoacán, Mexico

Danna Betsabe Rivera Ramírez[1], Cisteil X. Pérez-Hernández[2,3], Yaayé Arellanes-Cancino[4] and Luis Mendoza-Cuenca[2]

[1] Unidad Lerma, Universidad Autónoma Metropolitana, Lerma de Villada, State of Mexico, Mexico
[2] Faculty of Biology, Behavioral Ecology Laboratory, Universidad Michoacana de San Nicolás de Hidalgo, Morelia, Michoacán, Mexico
[3] IUCN SSC Firefly Specialist Group, Gland, Switzerland
[4] Facultad de Economía, Universidad Michoacana de San Nicolás de Hidalgo, Morelia, Michoacán, Mexico

Corresponding authors
Cisteil X. Pérez-Hernández,
cxinum@gmail.com
Luis Mendoza-Cuenca,
lfmendoza@umich.mx

## ABSTRACT

**Background:** Nocturnal fireflies are insects easily recognizable by their notable bioluminescence. They are also bioindicators of ecosystem health due to their sensitivity to environmental changes. In this study we employ oral history regarding fireflies and their habitats to compile the collective memory of the inhabitants of Morelia, in central-western Mexico, to analyze changes in biodiversity associated with urbanization.

**Methods:** The main tools we used were interviews and surveys, in addition to data from scientific literature, entomological collections and citizen science platforms. We explored fireflies as useful elements both to collect oral histories from volunteers (experts or non-experts on the topic) and to serve as a source of biological data (*e.g.*, current and past distribution of fireflies in the city, estimates of biodiversity loss, and threat factors), and even to analyze the potential loss of local ecological knowledge among human generations.

**Results:** A total of 112 surveys and interviews were conducted with people of three different generations and from different parts of Morelia to collect human demographic data, and spatial, temporal, abundance, and perception data on fireflies. We found local recognition of fireflies by most Morelians, as well as reports of a decrease in both the frequency and abundance of fireflies, and even identified sites of extirpation, *i.e.*, the disappearance of these insects from the environments where people used to see them. Morelians associated these phenomena with increased anthropogenic activities in the city, such as urban growth and a notable increase in pollution and deforestation. Most Morelians believe that the current conditions of the city are unsuitable for the existence of fireflies, and that it is important to conserve these insects because they play an important role in ecosystems and are of high environmental and aesthetic value. In contrast, the younger generation of Morelians showed lower interaction and recognition of these insects in nature, which could be related to the loss of collective memory over generations and the shifting baseline syndrome.

**Conclusions:** In contexts where historical scientific data are not available, we suggest that fireflies can be used to assess the history of natural environments and changes in the populations of these insects. Moreover, fireflies can be beneficial in terms of engaging people in conservation strategies, citizen science, and science communication.

# INTRODUCTION

Fireflies (Coleoptera, Lampyridae) are beetles that are characterized by emitting their own light (*i.e.*, they are bioluminescent). This group of insects has a wide range of geographic distribution and high morphological and ecological diversity (*Branham, 2010*). Fireflies contribute significantly to ecological networks and ecosystem stability and, due to their sensitivity to environmental change, they are also considered bioindicators of environmental integrity (*Hagen et al., 2015*; *Idris et al., 2021*). Their larvae are predators of snails, slugs, and earthworms (*Fallon et al., 2019*; *Riley, Rosa & Silveira, 2021*), and some species are used as biological control agents for snails associated with crops (*Fu & Meyer-Rochow, 2013*). Adult fireflies are a food resource for birds, lizards, bats, and spiders (*Lloyd, 1973*). From an anthropocentric point of view, these insects are also important because their bioluminescence is very striking to the human eye, and they have been integrated into human stories and culture for thousands of years as a result (*Ineichen, 2016*).

Adult fireflies typically exhibit annual activity cycles and in tropical regions present higher species richness and abundance during the rainy season, coinciding thus with the maturing or harvesting of native and cultivated plants and contributing to their cultural resonance (*Zaragoza-Caballero, 2004*; *Ineichen, 2016*). In some firefly species, adult males and females congregate in large numbers at the same site and fly through vegetation or over open fields while performing their bioluminescent courtship dances. The glow or flashes they emit to send their courtship messages vary widely in color, intensity, and intermittency, and some species synchronize their flashes in large flickering clouds which creates a great visual impact for the observer (*e.g.*, *Photinus palaciosi* in Tlaxcala, Mexico, and *Pteroptyx tener*, in Selangor, Malaysia; *Owens et al., 2022*). It is therefore common for these nocturnal displays of fireflies to create a lasting impression on their observers for decades.

Recent studies indicate that urbanization is one of the main factors that threatens the global insect fauna, especially fireflies, due to the loss of their habitats and other changes in the natural environment (*Jusoh & Hashim, 2012*; *Lewis et al., 2020*; *Wagner et al., 2021*). Moreover, increasing water pollution, artificial night light pollution, and pesticide use in recent decades have significantly impacted firefly and other insect populations (*Isenring, 2010*; *Brühl & Zaller, 2019*; *Wang, Cao & Wang, 2022*; *Vaz et al., 2021*; *Owens et al., 2022*). This phenomenon has been a challenge to document at regional and local scales in countries such as Mexico, unlike other countries such as England, where monitoring and

studies of insect diversity and abundance have been conducted for decades or even centuries (*e.g.*, *Gardiner & Didham, 2020*; *Tso et al., 2021*). In most Latin American countries, data pertaining to a "zero" or "previous" state of nature are scarce. However, these approaches are vital to recover as much data as possible to facilitate the creation of conservation and restoration strategies for the future, particularly in regions with the highest animal (including firefly) diversity and abundance. Citizen or community science may be a useful tool to address this issue since it encompasses the activities of the non-expert public in scientific discovery and research, data collection, monitoring, and experimentation across a wide range of disciplines, including environmental and ecological science (*Theobald et al., 2015*; *Fraisl et al., 2022*). It could also be useful to document the environmental history of developing cities.

Drastic changes in ecosystems also seem to be related to the loss of collective memory in human communities regarding past conditions of natural environments and habituation to existing environmental degradation (*Pauly, 1995*; *Soga & Gaston, 2018*). This psychological and sociological phenomenon is known as the shifting baseline syndrome and has been studied mainly in fisheries and in avian and tree communities (*Pauly, 1995*; *Papworth et al., 2009*; *Jöhnsson, Mårald & Lundmark, 2021*; *Soga & Gaston, 2018*).

In the city of Morelia in Michoacán, central-western Mexico, the process of transformation and expansion of urban zones that has taken place since the 1960s decade is the main cause of land cover change and biodiversity loss in the region (*López et al., 2001*; *MacGregor-Fors, 2010*; *Bollo Manent, Morales & Martinez Serrano, 2022*). This urban growth has increased exponentially, due to the massive development of housing, commerce, industry, and tourism that has taken place at an accelerated rate and with little or no planning (*López et al., 2001*; *López Núñez & Pedraza Marrón, 2012*; *Bucio-Mendoza, Vieyra & Burgos, 2017*; *Bollo Manent, Morales & Martinez Serrano, 2022*). Morelia is a medium-sized, mid-sized or intermediary city with a population of approximately 600,000 people living in the urban zone, which connects important rural and urban areas (*United Cities and Local Governments, 2020*; *Bollo Manent, Morales & Martinez Serrano, 2022*; *IMPLAN, 2022*; *Ruiz-López et al., 2022*). This category of city presents the fastest growing urban areas, usually with a lack of planning, and currently hosts 20% of the world's population and more than a third of the total urban population (*United Nations, 2018*; *United Cities and Local Governments, 2020*).

Recently, *Pérez-Hernández et al. (2023)* found that urbanization in Morelia has affected firefly populations in the region. In particular, although there are a high number of firefly species (26) in and around the city, there are also several areas where these insects are known to be extirpated. In their study, these authors utilized citizen science activities to obtain data on extirpations in Morelia that were crucial to understanding the factors negatively impacting fireflies and biodiversity in the city.

To evaluate whether oral histories could effectively gather data on firefly declines and test for shifting baseline syndrome among Morelian people, in this study, we used different tools from oral history to compile information regarding different aspects of firefly populations from Morelia; and to create collective representations and knowledge about the past state of the firefly populations and their habitats, as well as to document the

processes of change in the natural environment of the city, through the knowledge and experience of its citizens. As fireflies are usually recognized as common insects, we expected that Morelians would be able to recognize the term "luciérnagas" (fireflies in Spanish) regardless of their generation and that they would do so because of the characteristic flash of these insects. Since urbanization could led different childhood experiences with nature across generations (*Soga & Gaston, 2018*) and that some studies suggest diminishing in both fireflies abundance and occurrence throughout the recent decades at a global scale (*Lewis et al., 2020*); in a more local scale (Morelia city) we expected generational differences in how people know about fireflies and in their perceptions of firefly populations and habitats, as well as variations in the value placed on fireflies across generations. We also expected to identify historical firefly localities where older interviewees used to sight fireflies and are now clearly no longer occupied.

From the data compiled we present an evaluation of the collective memory of different human generations about the fireflies in the current urbanized zone of Morelia. The main tools we used were interviews and surveys, in addition to data from scientific literature, entomological collections and citizen science platforms. We explored fireflies as useful elements both to collect oral histories from volunteers (experts or non-experts on the topic) and to serve as a source of biological data (*e.g.*, current and past distribution of fireflies in the city, estimates of biodiversity loss, and threat factors), and even to analyze the potential loss of local ecological knowledge among human generations.

## MATERIALS AND METHODS

### Study area

This study was conducted in the city of Morelia, Michoacán de Ocampo, in central-western Mexico, a medium-sized city with approximately 1 million inhabitants (60% of them living in the urban zone and the rest living in the rural zones) and a population density of 708.122 ind/km$^2$ (*IMPLAN, 2022*; *INEGI, 2022*). Morelia is located between 19°27′06″ and 19°50′12″N, −101°01′43″ and −101°30′32″W with an average elevation of 1,920 m asl (*INEGI, 1993*). During the pre-colonial era, swamps covered the area currently occupied by the city and the human population remained in the peripheral areas (*López Núñez & Pedraza Marrón, 2012*). Later, during the Spanish colonization (from the 16th to the 18th century), the swamp areas were replaced with homes and agricultural and livestock zones and the city expanded rapidly until the mid-20th century. More recently, from 1960 to 2020, the city expanded its size 15-fold, presenting an average urban expansion rate of 1.8 km$^2$ or 1.6% per year (Figs. 1A–1C; *López et al., 2001*, *López Núñez & Pedraza Marrón, 2012*; *INEGI, 2022*; *IMPLAN, 2022*).

### Oral history compilation

Oral histories were collected through individual interviews to reconstruct and transmit the history of Morelians (Fig. 2A). The use of surveys was also integrated as a more appropriate technique to facilitate the collection of information from participants with little time available and in very crowded sites (*Zamorski & Kurkowska-Budzan, 2009*). The

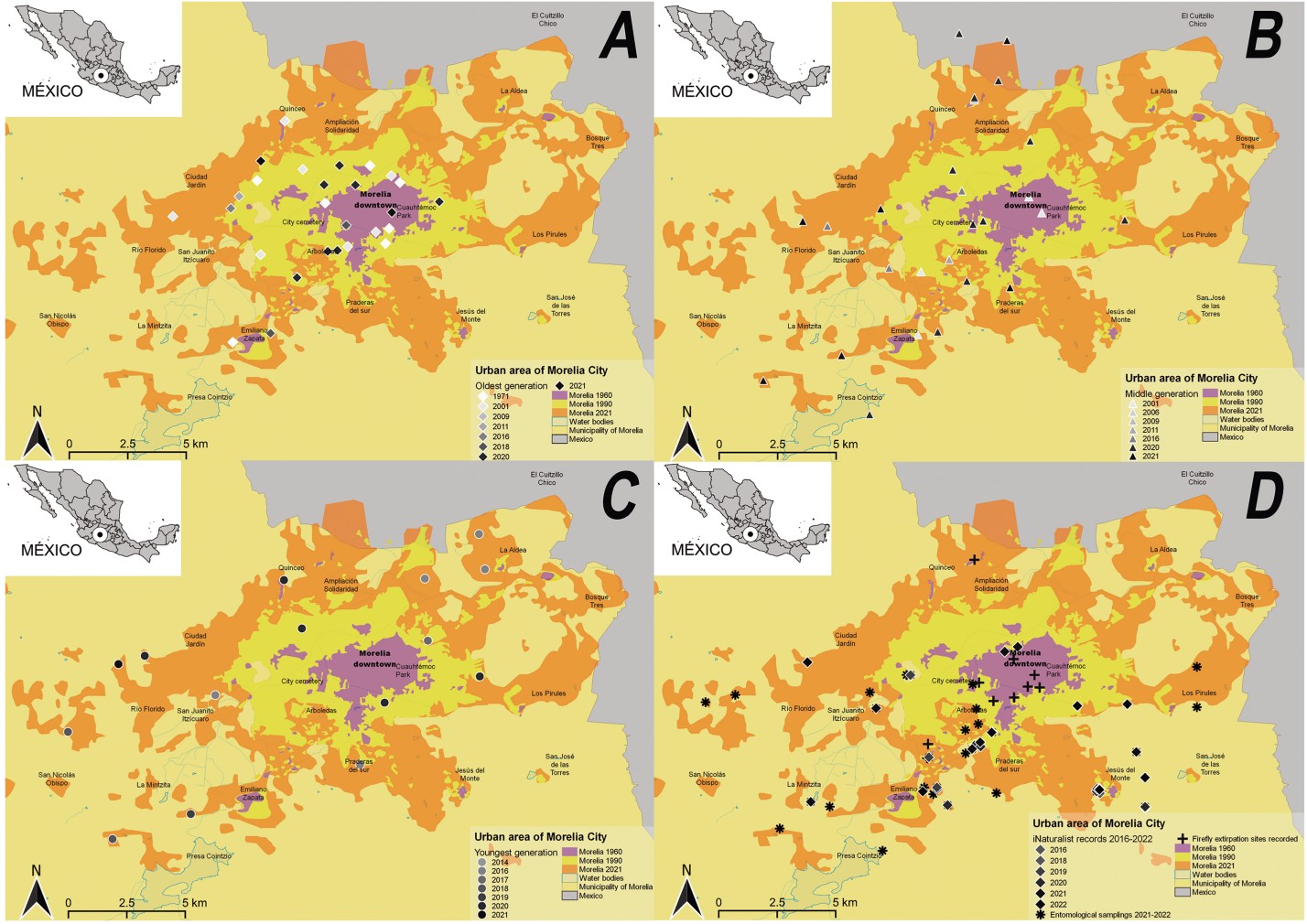

**Figure 1 Growth of the urban area of the city of Morelia (Michoacán, Mexico) in 1960, 1990, and 2021.** Based on *López et al. (2001)*, *López Núñez & Pedraza Marrón (2012)*, and *IMPLAN (2022)*. Sighting reports of firefly populations by (A) the oldest generation (rhombus), (B) the middle generation (triangles) and (C) the youngest generation (circles); (D) Verified occurrence records map (rhombus and asterisks) and firefly extirpation sites (black crosses; Table S1 and *Pérez-Hernández et al., 2023*). Map data (*IMPLAN, 2022*; *INEGI, 2022*).

interviews and surveys were recorded in three formats: voice recording, video, and on paper, in order to facilitate the subsequent work of writing up and systematizing the data.

The interviewees were at least 5 years of age and from Morelia, or with at least 5 years of residence in the city (Fig. 2B). This served to assess the congruence of the oral history with the changes that had occurred in the environment. The interviews or surveys were conducted by three individuals between April and June 2022. The participants were interviewed in their homes, in locations with a high influx of people (*e.g.*, urban parks, science fairs, bookstores, *etc.*; Figs. 2B–2D), and in different areas of the city to ensure the greatest possible diversity of areas of the city and residents. The data were processed in a database to facilitate their classification. In the initial phase, the data were transcribed *verbatim* and subsequently categorized in terms of wording and incorporation of

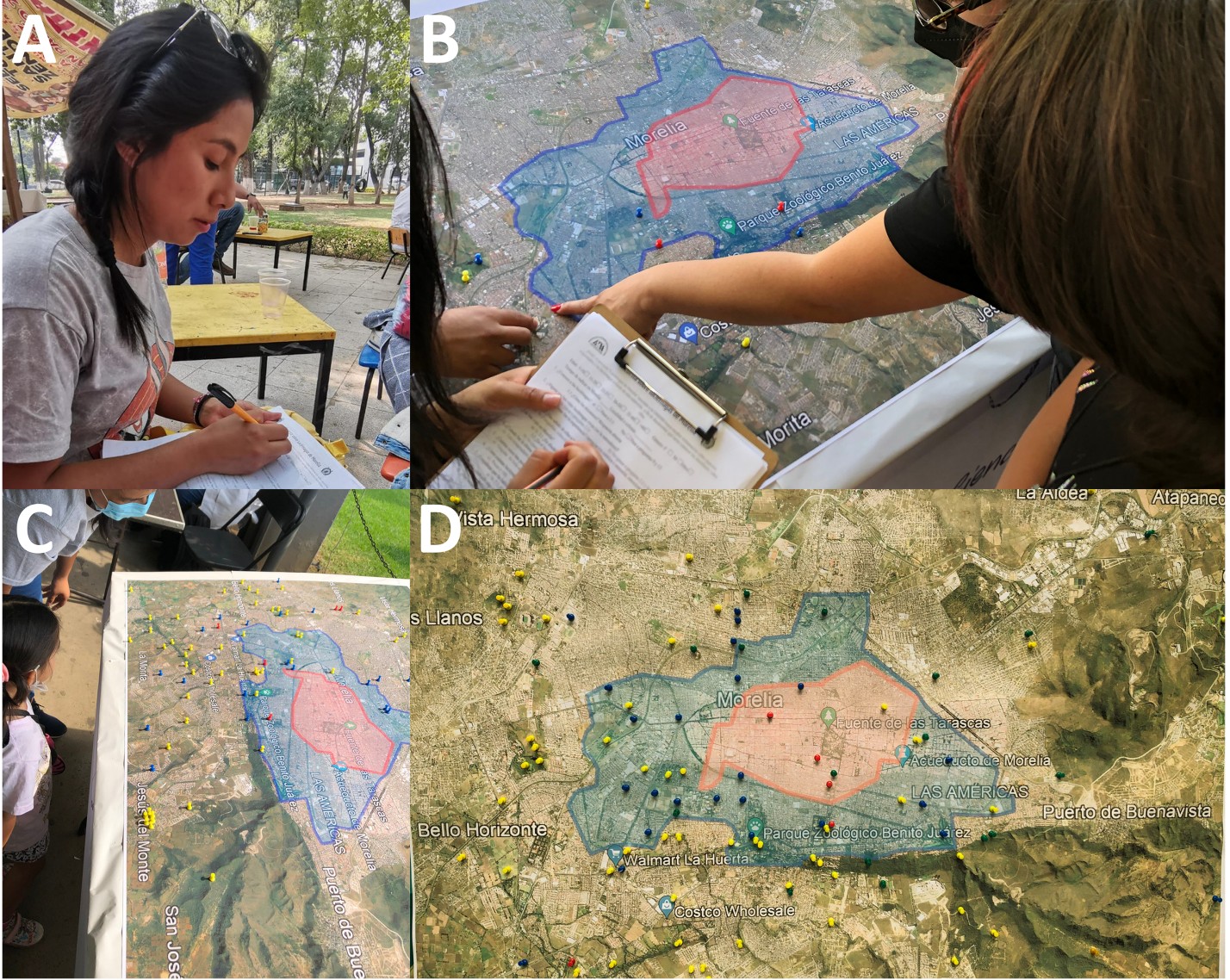

**Figure 2 Compilation of oral history on fireflies in Morelia city.** (A) Danna Rivera during an interview in 2021; (B–D) photos of the printed map of Morelia city illustrating the expansion of the urban area over the last 60 years and used during the interviews and surveys on fireflies in a science exhibition in the Michoacana University. Image credits: (A) Danna Rivera, (B–D) Cisteil X. Pérez-Hernández. Map data ©2021 Google, INEGI.

information from the testimonies for the purpose of analyzing the various documented narratives. The responses were grouped, simplified and categorized according to the options presented in the survey format or those that showed similarities, even when they came from different informants.

We employed an interdisciplinary methodology, integrating social and ecological science methods to understand how environmental and biodiversity changes are perceived by diverse communities across firefly habitats. This approach allowed us to potentially

assess the changes in firefly populations and the impact of human activities on this group of insects.

## Ethical statement

Ethical approval for this research was obtained from the "Committee of Bioethics, Ethics in investigation, investigation, and Biosecurity" and by the General Coordinator of the postgraduate programs of the Michoacana University of San Nicolás de Hidalgo to carry out this study. Consent for conducting the interviews and surveys was obtained verbally from all participants, who were fully informed about the purposes of our research and how their responses would be used and stored. Furthermore, no personal, biometric, or identity data was requested from those who voluntarily completed the survey.

## Interview structure

To collect the oral history, we employed the methodology proposed by Lara & Antúnez (2014), which consists of triangulation among questions, dialogue, and perception. To achieve this, we used a guide comprising (i) a brief presentation of the project, (ii) a section on the demographic and residency data of the participants, and (iii) questions on the spatial, temporal, abundance, and perception aspects of fireflies (Table 1). The presentation facilitated an atmosphere of trust and introduced the participants to the context of the project, without generating pressure or bias regarding the perceived importance of their testimony or responses.

The demographic data (age and sex) of the participants made it possible to recognize the participation of three generations; however, the variable *sex* was removed as a variable of interest because we did not find any significant relationship with the analyzed dependent variables. Regarding the generation, the older corresponds to those born in 1973 (now 51 years old) or before that year, the middle to those born between 1974 and 2001 (21 to 50 years old) and the younger to those born between 2002 and 2019 (5 to 20 years old). That categorization mainly corresponds to generations living their childhoods in moments of high rates in the urbanization processes and growth population in Morelia (see INEGI, 1993; López Núñez & Pedraza Marrón, 2012). The categorization not only allowed us to analyze discrepancies in the knowledge of fireflies among different generations of Morelians, but also to determine how each of these generations perceives the changes in the city and to evaluate the possible existence of a shifting baseline syndrome (SBS) in the population of Morelia. The participant's place of residence allowed us to cover the greatest possible diversity of areas within the city, thus avoiding bias in the data based on the history of any particular area.

The third section comprised 14 questions pertaining to data on past and/or current observations, or the lack thereof, of fireflies by Morelians. The objective was to obtain spatial, temporal, abundance, and perception data related to these bioluminescent insects (Table 1). We expected that Morelians would be able to recognize the term "luciérnagas" (fireflies in Spanish) regardless of their generation and that they associate these insects with

**Table 1 Questions applied to 112 inhabitants of the city of Morelia, in Michoacán, Mexico.**

| Question | Value range | Justification | n |
|---|---|---|---|
| 1. Do you know about fireflies?<br>1.1 How do you recognize them? | a) Yes<br>b) No<br>c) Not sure<br>    1.1 Open-ended question | The aim was to assess the general recognition of fireflies and to detect biases in the testimonies (*e.g.*, observations referring to cocuyos *i.e.*, glowing click beetles (Pyrophorini, Elateridae), or to "dragonflies", which are often confused with "fireflies" in Spanish). | 112 |
| 2. Have you ever seen fireflies in nature?<br>2.1 How was your experience? (aesthetic, sensorial, emotional appreciation)<br>2.2 Do you know them through any other media? | a) Yes<br>b) No<br>    2.1 Open-ended question<br>    2.2<br>a) Yes<br>b) No<br>c) Not sure | It is possible that the experiences between the interviewees and the fireflies may evoke a sense of rootedness, thereby imprinting themselves upon the memory. This question sought to elicit more detailed recollections of these experiences. | 109 |
| 3. Have you seen fireflies in Morelia? Where? | a) Yes<br>b) No | The aim was to geographically delimit the areas where fireflies have been present throughout the history of the city (since 1950). | 99 |
| 4. When was the last time you saw them in Morelia? (approximate time) | a) Last season (2021)<br>b) 5 years ago<br>c) 10 years ago<br>d) 15 years ago<br>e) >20 years ago<br>f) >50 years ago | By comparing the location and timing of sightings with environmental changes, we sought to document the history of possible environmental deterioration in the firefly habitats. | 72 |
| 5. How many times have you been able to see them in Morelia in the last few years? | a) Never seen again<br>b) Only once<br>c) Only a few times<br>d) Sometimes<br>e) Frequently | The testimony on the recurrence of sightings allowed us to obtain information on temporal patterns of firefly presence, and to associate these patterns with the environmental changes evaluated. | 72 |
| 6. How many fireflies do you think you saw the last time (per year or season)? | a) Few (<10)<br>b) Some (ca. 20–30)<br>c) Many (>50) | This was used to estimate the change in the size of past and current firefly populations. | 72 |
| 7. Do you think that there have been changes in the number of fireflies you have seen?<br>7.1 What are the changes? | a) Yes. 7.1 What are the changes? (Open response)<br>b) No<br>c) Doesn't know | This was used to document the perception of changes in the size of firefly populations. The objective was to document evidence of the gradual decline of firefly populations in the area. Knowledge about the historic and current status of fireflies was key to the interpretation of the shifting baseline syndrome (SBS) (individual and collective memory). | 62 |
| 8. Has anything changed in the places in Morelia where you once saw fireflies?<br>8.1 What changes? | a) Nothing has changed<br>b) Plantations<br>c) Constructions<br>d) Crops<br>e) Others | This information was used to detect the factors associated with the potential loss or diminution of the firefly distribution area, and to reconstruct the citizen perception of the increase in urbanization in Morelia. | 72 |

| Question | Value range | Justification | n |
|---|---|---|---|
| 9. Have you noticed changes in Morelia over recent years?<br>9.1 What changes? | a) Yes.<br>    9.1 Open response<br>b) No<br>c) Don't know | The aim was to gather information on the perception of the processes of change that have occurred in different areas of the city of Morelia. | 112 |
| 10. Do you think there should be fireflies in the town where you live?<br>10.1 Why? | Open-ended question | This allowed evaluation of the people's perception of the ecological requirements of fireflies as well as the characteristics of their natural environment. | 89 |
| 11. Do you think fireflies are important for the environment/ecosystems?<br>11.1 Why? | Open-ended question | This perspective offered an insight into the perception of fireflies in the Morelia area and the extent of current knowledge about them. | 109 |
| 12. Do you consider fireflies to be important for humans? | a) Yes. 12.1 Why? (Open response)<br>b) No. 12.2 Why? (Open response)<br>c) Don't know | Perception at the individual level forms part of the manner in which information is transmitted for the construction of a collective memory. | 109 |
| 13. Where you live have you ever heard stories about firefly sightings?<br>13.1 What stories have you heard?<br>13.2 Who told them to you?<br>13.3 From how long ago are these stories? | 13. a) Yes; then:<br>    13.1 Open ended question<br>    13.2 Open ended question<br>    13.3<br>a) Approximately 5 years<br>b) Approximately 10 years<br>c) Approximately 15 years<br>d) Approximately 20 years<br>e) More than 20 years<br>    13. b) No | Given that shared oral histories generate collective memory, this information was useful to estimate the manner and frequency of the transmission of oral histories regarding the past and current status of firefly populations. With this information, it was possible to evaluate the existence or loss of collective memory among generations. | 109 |
| 14. Do you often tell your stories about fireflies to others in your family or community? | a) Yes<br>b) No | This made it possible to evaluate the practice of oral history among the inhabitants of Morelia, and to establish the scope of oral histories in terms of generating a collective memory about the state of firefly populations and the natural environment. | 99 |

**Note:**
The participants came from three different generations and the aim was to evaluate their perception of fireflies and their natural environment, and to reconstruct the history of firefly populations in recent decades; n, number of participants who answered the question.

their flash (Question 1). We also expected generational differences in how Morelians know about fireflies (Questions 2 and 3 on sightings in nature and/or other media) and in their perceptions of firefly populations (Questions 5, 6 and 7 on abundance and frequency of firefly sightings) and habitats (Questions 8, 9 and 10 on the human perception of change in firefly habitats and its causes), as well as variations in the value placed on fireflies across generations (Questions 11 and 12). We also expected to document local extirpation of firefly populations based on the identification of historical firefly localities where older interviewees used to see fireflies and are now clearly no longer occupied (Questions 3 and 4), as well as to identify elements that indicate the existence of potential shifting baseline

syndrome among Morelian people (mainly questions 13 and 14 along the analysis of the complete survey).

## Analysis of change in firefly habitats and populations

Due to the absence of long-term monitoring or entomological data regarding historical firefly populations and habitats in Morelia, testimonies (sighting reports) from elderly and middle-aged citizens were gathered to identify past sighting sites. This information established a baseline of the potential distribution of fireflies in recent decades. The credibility and reliability of the obtained data came from the details shared by the participants and from the repetition and consistence of the information shared by the entire interview group; in addition, following the method of *Lara & Antúnez (2014)*, we consulted old maps of the city to corroborate the stories. Current distribution data was also compiled through citizen testimonies of the three generations.

To estimate changes in the natural environment of the city of Morelia, we utilized the geographic location of sites reported as current or past habitats of fireflies in the city (*e.g.*, locality and/or street) by participating citizens, as well as the time of the reported sighting. To ensure the accuracy of the information collected, during the surveys and interviews we used a map of the city that illustrated the expansion of the urban area over the last 60 years (Figs. 2B–2D). We also used verified species occurrence records from the citizen science project "Luciérnagas de Michoacán" up to 2021 (*iNaturalist, 2024*), the entomological collection of the Universidad Michoacana de San Nicolás de Hidalgo, and entomological collections conducted in 2021 and 2022 (see the maps and data in *Pérez-Hernández et al. (2023)*, *Pérez-Hernández, Mendoza-Cuenca & Romo-Galicia (2023)*) to determine the current geographic distribution of firefly populations and whether those records coincide with the reported sightings in past decades. Given the high and accelerated urbanization of Morelia in recent decades, we expected a high frequency of firefly population absences in traditional firefly habitats reported by interviewees. All data were georeferenced through photointerpretation and a historical comparison of satellite images from 1960, 1990, and 2021, as provided by the Instituto Municipal de Planeación de Morelia (*IMPLAN, 2022*) and the QGis 3.22.1 software.

Based on our hypothesis, we tested for associations among Morelian generations and their (i) observations of fireflies in nature, (ii) experiences and emotions on firefly sightings, (iii) perceptions of changes in sightings frequency and abundance of fireflies; (iv) perceptions of changes in historical firefly habitats, and (v) perceptions of firefly values for ecosystems and humans. All statistical analyses were performed using Pearson's Chi-squared tests with R 4.1.2 (*R Core Team, 2021*).

## RESULTS

A total of six genera and 19 nocturnal firefly species were recorded (*i.e.*, verified species occurrence records) through sampling collections, entomological collection material and images from iNaturalist, in the current urbanized areas of Morelia and surrounding areas during the project "Luciérnagas de Michoacán" of which this work is a part (Fig. 3; *Pérez-*

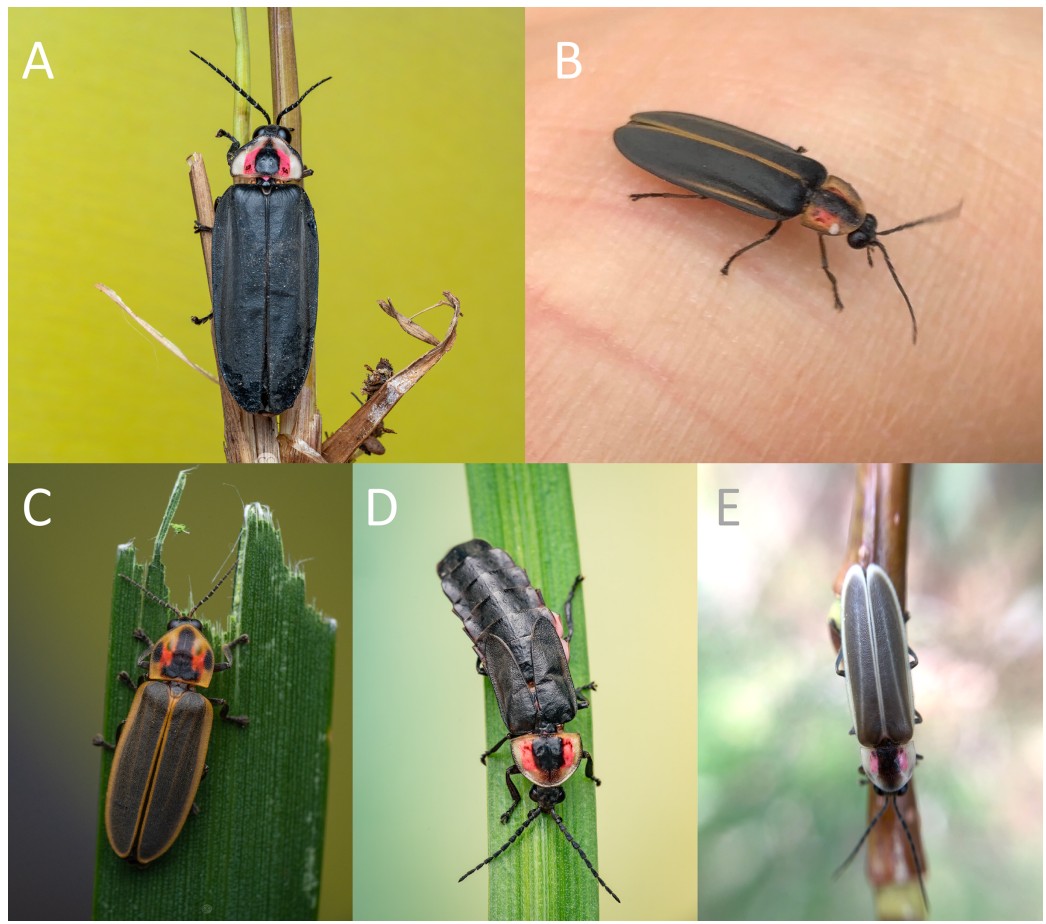

**Figure 3 Nocturnal fireflies (Coleoptera, Lampyridae) recorded in Morelia from 2016 to 2022.** The species were recorded using entomological samplings and citizen science platforms (*Pérez-Hernández, Mendoza-Cuenca & Romo-Galicia, 2023*; *iNaturalist, 2024*): (A) *Photinus barrerae* male; (B) *Photinus guillermodeltoroi* male; (C) *Pyractomena striatella* from La Mintzita, Morelia; (D) *Photinus extensus* female; (E) *Photinus acutiformis* from Universidad Latina de America intra-urban conservation area. Image credits: (A, C, D) Miguel Gerardo Ochoa Tovar; (B) Ek del-Val https://mexico.inaturalist.org/observations/14460873 CC BY NC; (E) Wendy Mendoza Arroyo https://mexico.inaturalist.org/observations/162239038, CC BY NC.               

*Hernández et al., 2023*; *Pérez-Hernández, Mendoza-Cuenca & Romo-Galicia, 2023*):
*Aspisomoides bilineatum* (Gorham, 1880), *Cratomorphus halffteri* Zaragoza-Caballero, 2012, *Photinus acutiformis* Zaragoza-Caballero and Cifuentes-Ruíz, 2023, *P. anisodrilus* Zaragoza-Caballero, 2007, *P. barrerae* Zaragoza-Caballero and Rodríguez-Mirón, 2023, *P. chipirietetsi* Zaragoza-Caballero and Vega-Badillo, 2023, *P. extensus* Gorham, 1881, *P. guillermodeltoroi* Zaragoza-Caballero and Rodríguez-Mirón, 2023, *P. leobonillai* Zaragoza-Caballero and Domínguez-León, 2023, *P. vegai* Zaragoza-Caballero and Cifuentes-Ruiz, 2020, *P. zuritai* Zaragoza-Caballero and Cifuentes-Ruiz, 2023, *Photinus* sp., *Pleotomus emmiltos* Zaragoza-Caballero, 2002, *Pl. pallens* Zaragoza-Caballero, 2002, *Photuris fulvipes* (Blanchard, 1846), *Ph. lugubris* Gorham, 1881, *Photuris* group *versicolor* (Fabricius, 1798), *Photuris* sp. and *Pyractomena striatella* Gorham, 1880. More specific

data on each record are available through GBIF.org, iNaturalist platform and can also be consulted in *Pérez-Hernández et al. (2023)*, *Pérez-Hernández, Mendoza-Cuenca & Romo-Galicia (2023)*.

Sighting reports were obtained from 112 participants who belonged to 84 (62.2%) of the total number of 135 neighborhoods registered in Morelia (Figs. 1A–1C, Datasets). No firefly species was identified by interviewees. Of the participants, 65 (58%) identified themselves as female, while the remaining 47 individuals (42%) identified as male. No participants identified with another gender category. The oldest generation of participants (individuals >51 years old) accounted for 32.1% (36 individuals) of the interviews, followed by the middle generation (21–50 years old) at 36.6% (41), and the youngest generation (5–20 years old) at 31.3% (35) (Datasets).

## Local knowledge and perceptions about fireflies

As we expected, most of the participants (97.3%, 109 individuals) indicated that they could recognize the term "luciérnagas" (fireflies in Spanish) and that they identify these insects by their bioluminescence. Furthermore, 88.4% had observed them in their natural habitat (Fig. 4A). Of these, 85.7% (96 individuals) indicated that their experience with fireflies was positive in those encounters (*e.g.*, excitement, admiration, awe), while the rest said it caused "fear" (Fig. 4B). Interestingly, of the total number of participants, only 2.7% (three individuals) said they did not recognize the term "luciérnagas", while 8.9% (exclusively members of the middle and the youngest generations) indicated that they were aware of fireflies through other media (*e.g.*, films, stories, photographs; Figs. 4A, 4B). As we expected there was a significant association regarding how different generations of Morelians knew about fireflies ($\chi^2$ = 6.882, *df* = 2, *p* = 0.032; Fig. 4A).

## Perspective of the inhabitants of Morelia regarding the loss of and threats to fireflies

At least 72 of the 109 participants (66%, belonging to all three generations), who had previously observed fireflies in their natural habitat, recalled seeing them in Morelia at different times: >50 years ago (8.3%), >20 years ago (12.5%), >10 years ago (6.9%), 5 years ago (20.8%), and 1 year prior to the survey (51.4%) (Fig. 5A). In each of the three generations, the highest proportion of memories of firefly sightings was reported in 2021; *i.e.*, in the most recent season (Fig. 5A).

The information obtained indicated a decline in firefly sightings in Morelia over the last decade. A total of 30.6% of the participants (*n* = 72) have ceased observing fireflies in their environment or have seen them only a few times (29.2%) or only once (13.8%). Only 19.3% of the participants continue to observe fireflies frequently (every rainy season over the past 10 years), while 7.1% do so occasionally (over the last decade) (Fig. 5B). In the oldest generation, the perception of firefly loss was high; a notable 57.7% of this generation reported a noticeable decline in firefly populations over time (Fig. 5B).

Regarding the perceived abundance of fireflies observed by the participants during each sighting in the last decade, 56.9% (*n* = 72) observed few fireflies, 11.1% saw a normal

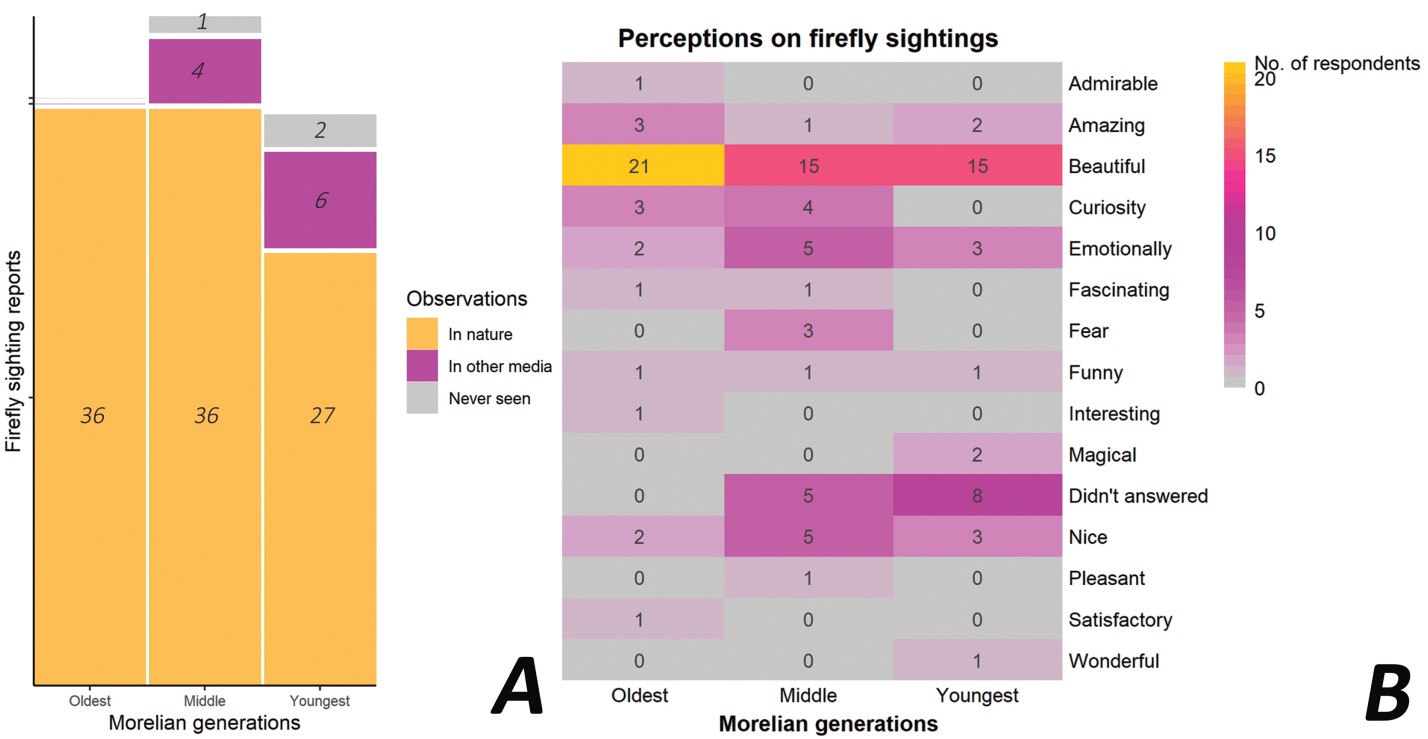

**Figure 4 Local knowledge and perceptions about fireflies in Morelia city, Michoacán, Mexico.** (A) Observations of fireflies in their natural habitat, other sources or have never been seen; and (B) experiences and emotions on firefly sightings among three different Morelian generations, from a total of 112 interviewed people.

number, and 32% indicated a high number. Of these, 32.7% belonged to the oldest generation, 34.4% to the middle generation, and 32.7% to the youngest generation (Fig. 5C). No significant differences were found among generations in their perceptions of changes in firefly abundances through time. However, most participants recalled that, in 2021 (pre-survey year), they observed the lowest abundance of fireflies compared to previous years. At least 73.6% (53 individuals) perceived a diminution or disappearance of fireflies from their usual sighting sites, while only 2.7% (two individuals) observed an increase in the number of fireflies compared to previous years (Fig. 5D).

A total of 22.2% of the participants (16 individuals, $n = 72$) indicated that there have been no changes in the site where they saw fireflies. Of these individuals, eight were from the youngest generation. In contrast, 85% of those over 20 years of age (from the oldest and middle generations) reported local changes in firefly habitats, including the loss or diminution of natural areas due to increased anthropogenic activities. There was a significant association among Morelian generations and the perceived changes in firefly habitats ($\chi^2 = 34.284$, $df = 14$, $p = 0.001$). Construction activities were identified as the main cause of the perceived environmental deterioration (61.1% of the participants, 40 individuals over 20 years old). Only 10.7% of the participants (12 individuals, $n = 112$) indicated that there have been no changes in Morelia in recent years, with most of these individuals belonging to the youngest generation (eight individuals). The remaining 89.3%
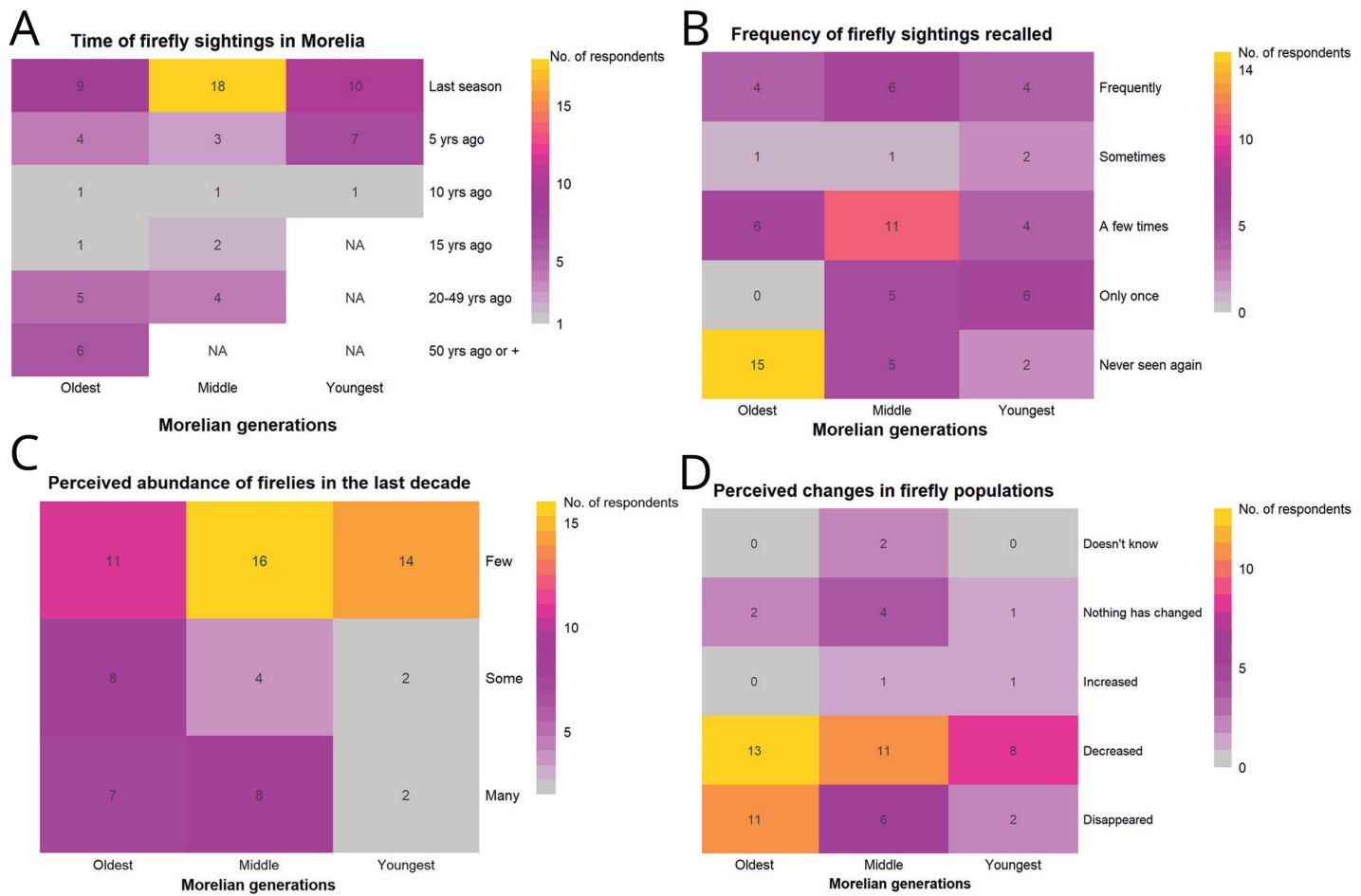

**Figure 5 Perceptions of Morelians regarding the changes in firefly populations.** (A) Observations of fireflies in different times among Morelian generations; (B) frequency of recalled firefly sightings by generation of interviewees during the last decade; (C) perceived abundance (individual number) of fireflies seen at the time of the sighting; (D) perceived changes in the abundance seen in the last decade.

(100 individuals) did perceive important changes in the city in recent years. These changes corresponded to growth in the urban area (89%), pollution (4%), deforestation (4%), and other factors (3%) and the perception of those changes were significantly associated with the different generations of Morelians ($\chi^2$ = 31.997, $df$ = 10, $p$ < 0.0004; Fig. 6A). Furthermore, 61.8% (55 individuals; $n$ = 89) considered that it was not realistic or likely to have the presence of fireflies in their locality. The majority (69.5%, 38 individuals) attributed this to the unsuitable current conditions in the city for these insects.

## Environmental and personal value of the fireflies for humans

The majority of participants who did recognize fireflies (78%, $n$ = 109) considered them an important component of ecosystems and are of high environmental value (Fig. 6B). Of these, 11.9% (13 individuals) attributed their importance to the role they play in food webs (*e.g.*, as predators of other animals and pests), 56.8% (62 individuals) attributed an intrinsic

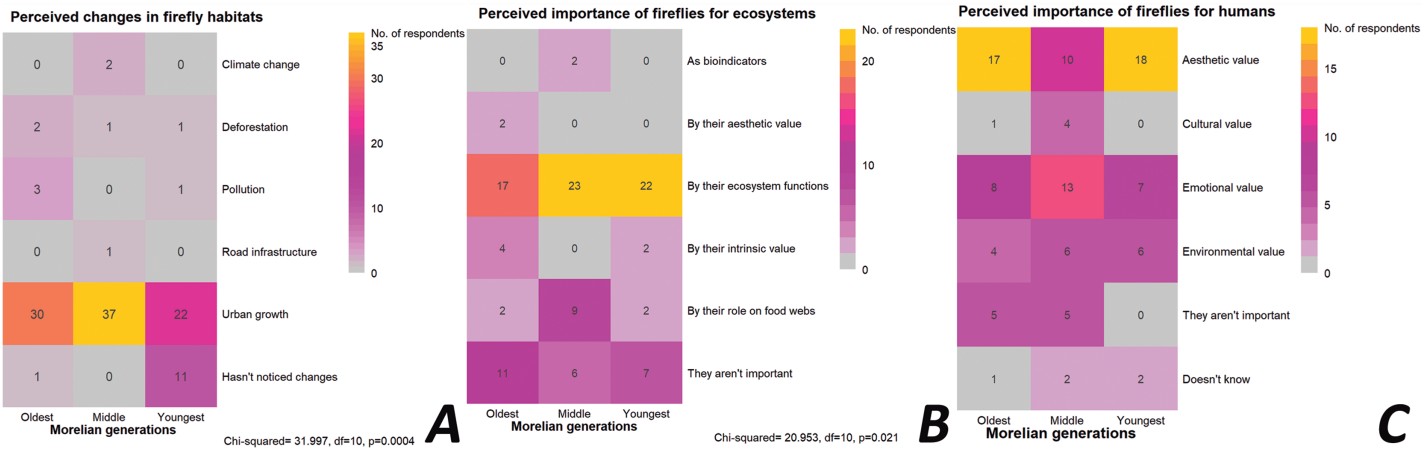

**Figure 6 Perceptions of Morelian people about habitat changes and value of fireflies.** (A) Changes in firefly habitats in the last years; (B) value of fireflies for ecosystems; (C) value of fireflies for humans.

value (*e.g.*, "all living beings fulfill a function in the environment, and that is why they continue to exist"), and 9.3% (10 individuals) considered that they have an aesthetic value (*e.g.*, they visually enrich the environment). The remaining 22% (24 individuals, $n = 109$) demonstrated no particular interest in fireflies and did not consider them to have an environmental value (45.8% of the oldest generation, 25% of the middle generation, and 29.2% of the youngest generation). A significant association was found between generations and their perceptions of the importance of fireflies to ecosystems ($\chi^2 = 20.953$, $df = 10$, $p = 0.021$).

A total of 9.2% of the participants (10 individuals, $n = 109$) did not consider fireflies to be important for humans, while 4.6% (five individuals) said they did not know (Fig. 6C). However, 86.2% of the participants (94 individuals) indicated that fireflies are important to humans for the following reasons: (i) their emotional value (29.8%), including the joy they bring when remembering and/or seeing them, or that they create greater empathy towards nature; (ii) their aesthetic value (47.9%) because they are "a delight to behold" or "a real spectacle"; (iii) their cultural value (5.3%) because these individuals consider that fireflies form an important part of the community and the identity of the people of Morelia; and (iv) their environmental value (17%) because fireflies are important for maintaining ecological balance and as a bioindicator species. There were no significant associations between generations and their perceptions of the importance of fireflies to humans.

**Transmission of collective memory among the inhabitants of Morelia**
Most participants (82.6%, $n = 109$) had never heard stories about these bioluminescent insects, either from family members or other members of their community. Only 17.4% (all belonging to the oldest and middle generations) related some anecdotes previously shared by family members. In total, 19 stories were obtained, dating back more than a decade, which described aspects of their lived experiences (42.1% of the stories) and the decline in the abundance and frequency of firefly populations (57.9% of the stories). For

example, their ancestors told them that "there used to be more fireflies", "before the constructions in the southern part of Morelia, there were a lot of fireflies", and "in my childhood, I saw a lot of fireflies". Furthermore, 34.4% of the participants ($n = 99$) indicated that they had never shared their experiences or anecdotes about fireflies with other individuals, while 65.6% had shared their stories with their family and friends.

**Historic and current distribution of fireflies in Morelia**

A total of 116 sites of nocturnal firefly occurrences were identified in Morelia. Of these, 84 (72.4%) were sightings reports by the interviewed participants (Figs. 1A–1C), and the rest were verified occurrence records: two (1.7%) from the entomological collection of the UMSNH; 15 (12.9%) from the *iNaturalist* platform (2016–2022; Fig. 1D), and 15 (12.9%) from entomological collections (Fig. 1D) of the "Luciérnagas de Michoacán" project (already published in GBIF.org and *Pérez-Hernández et al., 2023*; *Pérez-Hernández, Mendoza-Cuenca & Romo-Galicia, 2023*). The highest frequency of firefly reports (the compiled memories of sightings in the past) was found in the current intra-urban and peri-urban areas (Figs. 1A–1C). The most frequently reported sites across the different information sources and time periods were the Bosque Cuauhtémoc, Tenencia Emiliano Zapata and Jesús del Monte, where fireflies can still be seen today although in few numbers; except in the peri-urban Jesús del Monte where the populations can be considered as healthy. Only seven sites were identified from sources dating back over 50 years, 16 sites from 1994 to 2012, and 50 sites from 2016 to 2022 (Figs. 1A–1C). The sightings from the 1970s were concentrated in what was then the peri-urban zone (today the intra-urban zone) and only in one site (an urban park) is it still possible to see a few fireflies. From 1994 onwards, there were more sightings in the intra- and peri-urban zones, and less than half still have nocturnal fireflies.

## DISCUSSION

Our study is the first effort in Mexico to ascertain changes in biodiversity associated with urbanization over time, by employing citizen science and oral history data. The results covered a wide range of residential areas and ages of inhabitants of Morelia and demonstrated the local recognition of fireflies among Morelians. The data collected through the oral histories of the local inhabitants enabled us to document a decline in the populations of these insects over recent decades and to identify the factors associated with this decline. This would not have been achievable by consulting scientific publications or existing data sets alone, nor without the anecdotal experiences of the middle and oldest generations who provided testimony of fireflies decline and even local extirpation in specific sites where we physically surveyed the current absence of nocturnal fireflies (Fig. 1D; Table S1 and *Pérez-Hernández et al., 2023*). The results also provide evidence of a decline in interactions between the inhabitants of Morelia and fireflies, which reflects the decrease in the general interaction of citizens with nature over generations. Although the Morelian citizens placed a high personal and ecosystem value on the fireflies, we found no

evidence of community-level appreciation and there was a limited transmission of firefly-related stories across generations.

Our findings indicate that the inhabitants of Morelia readily identify fireflies by their bioluminescence and often associate positive emotions with these memories and experiences, as documented in other regions (*e.g.*, *Lewis et al., 2021*; *Lloyd, 1983*). In the case of individuals who indicated that they were unfamiliar with fireflies, it is possible that the interviewees use a different term for the insects than that mentioned during the interviews ("luciérnagas"). These could include terms such as "alumbrador", "lucecitas" and *chpíri etetsï* (meaning fireflies, in Purépecha). For this reason, we recommend that future studies employ a greater range of expressions to obtain a more diverse set of testimonies.

All individuals in the oldest generation reported having observed fireflies in their natural habitat. This is in contrast to the middle and the youngest generations, where a proportion of the participants only gained their knowledge of fireflies through other media (*e.g.*, films, photographs, social networks). Furthermore, the knowledge and experiences of the oldest generation of Morelians are strongly associated with the fact that these individuals had greater interaction with the natural environment. *Guerrero-Martínez (2015)* states that the manner in which people relate to their environment generates diverse notions regarding nature. Local views suggest that the rapid shift from a rural to an urban environment in Morelia over recent decades may be generating biases in the human-environment relationship, particularly among the middle and the youngest generations. In this sense, this study revealed that the observation sites of the older individuals were mainly in what had been the peri-urban and extra-urban areas at the time of observation, with a similar pattern evident for the middle generation. In contrast, most of the observations of the younger individuals occurred in the current urbanized area, which may reflect the fact that they no longer directly "experience nature", either because their neighborhoods no longer have green areas where they can carry out outdoor activities, or they do not usually engage in this type of activities. Therefore, we recommend that perceptions of the youngest people about environmental changes through time should be treated separately to avoid biases and/or take the shift in their perceptions into account when making general conclusions.

Remarkably, recent sightings (*i.e.*, 2016–2022) were associated with green areas located within the current intra-urban zone (*e.g.*, gardens, central reserves, parks, cemeteries). As evidenced in other studies, these areas help maintain insect populations in urbanized areas (*Raupp, Shrewsbury & Herms, 2010*) and are also the areas to which people of the youngest generation living in urbanized areas have greater access and are where these individuals can interact with nature. In particular, fireflies require sites with vegetation, high soil moisture levels, and low levels of light pollution (*Fallon et al., 2019*; *Idris et al., 2021*; *Pérez-Hernández et al., 2023*). However, in the current urbanized area of Morelia, public spaces that meet these requirements are scarce (*Bollo Manent, Morales & Martinez Serrano, 2022*). Only the peri-urban and extra-urban areas present the optimal conditions required for fireflies and other insects (*Pérez-Hernández et al., 2023*), as well as for interaction between humans and nature.

In the last 5 years, firefly sightings in Morelia and surrounding areas have been more numerous than in previous years, which we associate with the recent creation and dissemination of citizen science projects designed to record Mexican biodiversity (*e.g.*, *iNaturalist, 2024*; *Buscando Luciérnagas* project by *del-Val et al. (2024)*). However, the middle and oldest generations of Morelians also reported a notable decline in sightings compared to past seasons, with a continued reduction in frequency of sightings over recent years. This more recent decrease could be due to various factors associated with urbanization, including habitat loss, contamination of soils and water bodies, and light pollution (*Lewis et al., 2020*; *Fallon et al., 2021*; *Owens et al., 2022*; *Vaz et al., 2023*). These insects are also highly sensitive to variations in temperature and seasonality of rainfall, which have increased in recent years (*Evans et al., 2018*).

In the city of Morelia, firefly extirpation zones are associated with a low proportion of green spaces and high levels of light pollution (*Pérez-Hernández et al., 2023*), which can be added to the environmental deterioration of the city and particularly to the increase in urban area (*Bollo Manent, Morales & Martinez Serrano, 2022*). The inhabitants of this city have perceived these changes in their surroundings as negative. In particular, construction activities were identified as the main cause of the environmental deterioration of their natural surroundings, which is also associated with the demographic growth recorded in the area since the 1960s (*López et al., 2001*, *López Núñez & Pedraza Marrón, 2012*).

## Fireflies as tools to engage local communities in conservation efforts

The responses of the citizens of Morelia indicate that, for them, fireflies are of considerable importance in ecosystems and possess high environmental, intrinsic, and emotional value. The intrinsic value can be leveraged to highlight the importance of conservation and to modify our interaction with the surrounding environment (*Batavia & Nelson, 2017*). The high emotional value that the citizens of Morelia ascribe to the fireflies could be linked to social issues, such as sentimental attachment or nostalgia, *i.e.*, a deep emotional connection or bond that someone has to a place or experience. This could facilitate a more complex and profound understanding of the importance of generating awareness about non-human fauna (*Nghiem et al., 2021*; *Mata et al., 2021*).

In this study, we compared the opinions and experiences of different generations regarding the observation of fireflies and nature to analyze the environmental, social, and cultural changes that have taken place in the city of Morelia over time. As expected, we found that changes in reported firefly sighting sites and the gradual loss of fireflies were perceived mainly by individuals of more than 20 years in age (from the oldest and middle generations). This is because, having lived longer, these individuals possess more information, having experienced past conditions and being aware of the changes that their environment have undergone, compared to individuals of the youngest generation (*Guerrero-Gatica, Aliste & Simonetti, 2019*). Furthermore, the youngest generation demonstrated greater disinterest in fireflies and their significance. This could be due to the lack of information about these insects or a general lack of engagement with their natural environment and could act to generate biases in the collective memory and increase the detachment of the younger generations from their natural surroundings.

It has been suggested that the lack of intergenerational transmission of information, particularly orally, can lead to generational amnesia regarding changes in the natural environment (*Soga & Gaston, 2018*). In the case of the inhabitants of Morelia, this lack of oral history transmission could lead to the development of SBS. For example, the younger generations' perspective regarding their environment, which is severely degraded, as well as their minimal interaction with non-urbanized environments, could generate a lower expectation of what a healthy environment should look like (*Pauly, 1995*; *Soga & Gaston, 2018*). This situation could complicate the development of effective ecosystem conservation, restoration, and management strategies (*Humphries & Winemiller, 2009*; *Bonebrake et al., 2010*; *Bilney, 2014*; *Soga & Gaston, 2018*; *Guerrero-Gatica, Aliste & Simonetti, 2019*).

There has been considerable debate about the veracity and reliability of oral history as a source of academic information. This is due to the potential subjectivity of the information collected, and the impact of various factors (*e.g.*, social class, gender, territory, generation, culture, the "memory illusion" phenomenon) (*Ortíz, 2005*; *Papworth et al., 2009*). However, it has also been proposed that oral history, when transmitted by different people, can be shared and/or compatible with each other, such that the stories are related and serve to generate collective memories of the past (*Kansteiner, 2002*). Furthermore, distortions in the subjectivity of the stories can be reduced if they are analyzed according to the objectives, interests, and value system implicit in the research (*Barbieri, 2007*). In our study, we employed a variety of information sources to compare the testimonies and reinforce the data on extirpation sites of the firefly populations (*e.g.*, databases and maps of urban growth in Morelia). The interviews were also structured and categorized in a manner designed to systematize the information obtained. However, to better document the history of environmental changes based on oral history and anecdotes, we highly recommend that studies should focus on people older than 20 years to avoid potential biases in data interpretation.

Other studies have suggested the possibility of employing anecdotal evidence about the presence or absence of fireflies to document their extirpation and changes in populations (*Wong & Yeap, 2012*; *Lewis et al., 2020*; *Ridenhour, 2022*). Based on our findings, we also recommend the use of oral history about fireflies as a valuable tool for evaluating the history of some ecosystems and SBS (see also *Pérez-Hernández & Rivera-Ramírez, 2024 in press*). This is due to the following factors related to the fireflies: (i) they are insects that are easily recognizable by people due to their bioluminescence, (ii) they have a visual impact that can elicit diverse emotions and create a lasting impression in the memory (*Ineichen, 2016*), (iii) they form aggregations that differ sufficiently from other bioluminescent organisms (*e.g.*, Elateridae, Phengodidae), (iv) they have populations that were previously very common and abundant, and changes in their presence and quantity are perceptible (*e.g.*, *Lewis, 2016*; *Faust, 2017*; *Lloyd, 2018*; *Lewis et al., 2020*), (v) many species emerge at specific times of the year (*e.g.*, rainy season, *Zaragoza-Caballero, 2004*), and it is therefore possible to obtain information about people's attention to seasonal phenomena and their environment (*e.g.*, *Juárez Becerril, 2017*, *2022*; *Takada, 2012*; *Ineichen, 2016*), (vi) people associate them with well-conserved habitats, so information about their perception of

historic changes in their environments and their appreciation of nature can be compiled (*Badillo, 2016*), (vii) the systematic and objective collection of oral histories regarding fireflies, and their comparison with other sources of information, can provide direct evidence of the intensity, causes, and possible consequences of the extirpation of fireflies, (viii) accessing the memories held by people regarding the fireflies can generate information about the transmission of oral stories across generations, which can be used to stimulate collective memory and reduce generational environmental amnesia, (ix) information from oral stories could be used to assess the cultural impact of the extirpation or extinction of insects such as fireflies, and (x) documenting and reporting the loss of these charismatic insects and its causes could serve as a catalyst for increased citizen participation in conservation and sustainable management strategies for ecosystems (*Lewis et al., 2021*).

## CONCLUSIONS

Our findings suggest that different generations of inhabitants of the city of Morelia have perceived a decline in firefly populations which they associate with the loss of green areas and the expansion of urban infrastructure in the city. This information is invaluable for assessing and monitoring firefly diversity in Morelia, including identifying extirpation sites. The compilation of oral history also allows us to preserve local knowledge and expand the research landscape in a context of scarce sources of historic scientific information. It also enables evaluation of the shifting baseline syndrome present among the citizens of Morelia. This is an area of significant interest for conservation efforts, including those focused on fireflies and other species in Morelia, which are conducted by disseminating knowledge about the species and the threats they face. Despite the challenges involved in using oral histories, they can provide valuable insights into historical events that are often overlooked but still have a significant impact on current perceptions of different socio-environmental problems. The methodology and findings of our study could be applied to other medium-sized cities with a similar history of urban development. This will enable a deeper understanding of their history of environmental deterioration and its causes and its consequences. Finally, given the charisma ascribed to fireflies by humans, they could be utilized as flagship species in citizen science projects for the study and monitoring of biodiversity, as well as in the generation and implementation of conservation and sustainable management projects for the natural environment.

## ACKNOWLEDGEMENTS

Cisteil X Pérez-Hernández thanks Olivia Huerta Luna, Ana María Gutiérrez Mancillas and Yuritzi Román Garibay, for their great assistance in the interviews and surveys in 2022. Special thanks to David Venegas Suárez Peredo and Ek del Val for their valuable suggestions and discussions on oral history techniques and survey questions respectively. We are grateful for the "Tianguis de la Ciencia" annual event for bringing these science projects to the public, as well as all the people who provided their time and shared anecdotes on their experiences with Morelian fireflies. We also thank the iNaturalist citizen

participation and Javier Ponce Saavedra for providing entomological material. We are grateful to Daniel Silva, Richard Joyce and Christopher Paradise who made valuable comments and suggestions that significantly improved this article. Our manuscript was proofread by professional English translator Keith MacMillan.

### Funding

Interviews and surveys for this work were funded by the Coordination of Scientific Research, Universidad Michoacana de San Nicolás de Hidalgo (UMSNH) and the Institute of Science, Technology and Innovation of the Government of Michoacán, through the Program for Scientific Research Projects of Regional Impact (PICIR-071). The National Council of Science and Technology (CONAHCyT) funded Cisteil Xinum Pérez-Hernández's postdoctoral project (CVU 365258), of which this publication is a part, through the program "Estancias Posdoctorales por México". This work was published using the annual institutional membership of the Universidad Michoacana de San Nicolás de Hidalgo through the joint support of the Institute of Science, Technology and Innovation of the Government of Michoacán, through the Program "Comparte tus Ideas" (ICTI/D.A./213/2023) and the Coordination of Scientific Research, Universidad Michoacana de San Nicolás de Hidalgo. There was no additional external funding received for this study. The funders had no role in study design, data collection and analysis, decision to publish, or preparation of the manuscript.

### Grant Disclosures

The following grant information was disclosed by the authors:
Coordination of Scientific Research, Universidad Michoacana de San Nicolás de Hidalgo (UMSNH).
Institute of Science, Technology and Innovation of the Government of Michoacán.
Program for Scientific Research Projects of Regional Impact: PICIR-071.
The National Council of Science and Technology (CONAHCyT): CVU 365258.
Estancias Posdoctorales por México.
Annual institutional membership of the Universidad Michoacana de San Nicolás de Hidalgo through the joint support of the Institute of Science, Technology and Innovation of the Government of Michoacán, through the Program "Comparte tus Ideas": ICTI/D.A./213/2023.
Coordination of Scientific Research, Universidad Michoacana de San Nicolás de Hidalgo.

### Competing Interests

The authors declare that they have no competing interests. Cisteil X. Pérez-Hernández is a volunteer expert of the IUCN SSC Firefly Specialist Group.

## Author Contributions

- Danna Betsabe Rivera Ramírez conceived and designed the experiments, performed the experiments, analyzed the data, prepared figures and/or tables, authored or reviewed drafts of the article, and approved the final draft.
- Cisteil X Pérez-Hernández conceived and designed the experiments, performed the experiments, analyzed the data, prepared figures and/or tables, authored or reviewed drafts of the article, and approved the final draft.
- Yaayé Arellanes-Cancino conceived and designed the experiments, analyzed the data, authored or reviewed drafts of the article, and approved the final draft.
- Luis Mendoza-Cuenca conceived and designed the experiments, analyzed the data, authored or reviewed drafts of the article, and approved the final draft.

## Human Ethics

The following information was supplied relating to ethical approvals (*i.e.*, approving body and any reference numbers):

The Committee of bioethics, Ethics in investigation, investigation and Biosecurity of the Universidad Michoacana de San Nicolás de Hidalgo certified the Ethical approval to carry out the study in Morelia, Michoacán, México.

## Data Availability

The raw data of interviews and surveys are available in the Supplemental Files.

## Supplemental Information

Supplemental information for this article can be found online at http://dx.doi.org/10.7717/peerj.19413#supplemental-information.

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
