# Peer review of "Oral history as a citizen science tool to understand biodiversity loss and environmental changes: on firefly extirpation in Morelia, Michoacán, Mexico"

_PeerJ, doi:10.7717/peerj.19413_

## Round 0.1 · original submission · Minor Revisions

Dear Dr. Pérez-Hernández,

After this first review round, both reviewers found your study interesting and worth to be published in PeerJ. Considering both positive positions, I believe your manuscript is almost ready for publication in the journal. Still, minor improvements are still required before the text is formally accepted for publication. I believe the suggestions made by both reviewers will greatly improve the quality of the study.

Congratulations on your hard work regarding this topic of study!

Sincerely,
Daniel Silva

·

Basic reporting

In general, the article is written clearly and there are minimal issues with the English. I have suggested a few changes to word choice as comments in the pdf to reduce ambiguity in a few cases. For example, "extirpation" may be a more appropriate word in this context than "extinction" (lines and "firefly occurrence sites." I also recommend referring to the generations by their relative age (younger, middle, older) rather than by "first," "second," and "third."

The introduction frames the relevance of this study in terms of firefly conservation, data gaps that make it difficult to document declines and extirpations, and the phenomenon of shifting baseline syndrome. I found the literature review to be generally comprehensive and adequate. My main suggestion for the framing of the study would be to be more specific and explicit about the questions being asked, particularly those analyzed with statistics.

This study did not present hypotheses (a priori or post priori), but rather several areas of inquiry and exploration. Given the interdisciplinary nature of the study, this was not necessarily a problem, but it did make it slightly less clear to the reader which results were most salient or important.

While the raw data for the survey and interview results were shared in English, I would recommend providing a copy in the original Spanish as well, if allowed by PeerJ. I suggest this because survey responses may be influenced by subtleties of the wording of questions.

Experimental design

I found the study's application of social science tools (oral history interviews and surveys) to the topic of firefly declines in the face of increasing urbanization to be compelling and meaningful. The authors clearly communicate how this study contributes to an improved understanding of both trends in Morelia's firefly populations over time and the potential of oral histories and social science methods to gather information that would be difficult or impossible to collect using the traditional methods of conservation biology.

The survey and interview methods were described clearly and completely. I think that more detail could be provided about how other data types (specimen records and citizen science occurrence records) were used or analyzed, as well as how they were integrated with the social science data.

No hypotheses are stated up front, nor are test variables explicitly named. Even if there were not a priori hypotheses, I would recommend framing methods and results in terms of expected results given possible phenomena. For example, if the shifting baseline syndrome were at play, it might be expected for older study participants/respondents to perceive declines in fireflies, while participants of younger generations would not note or describe declines. If local extirpations are assumed, one would also expect older participants to make note of historic firefly localities that are now clearly no longer occupied (either confirmed via surveys or inferred based on severe land-use change).

Validity of the findings

As a reader, I took away from this study a sense that oral histories and social science methods show promise tools for characterizing perceptions the status and trends of firefly populations, as well as revealing how and why community members might place value on fireflies as organisms.

The authors mention that data from specimen collections, citizen science platforms and previous field studies were used in this study. However, it is not clear to me how these data were used exactly, and these data were not provided.

Figure 1 shows a map of Morelia with reported firefly sighting localities symbolized by the generation of the observer. It is unclear to me whether the other (non-interview or survey-generated data points) are also featured on this map. While the generation of the observer is an interesting variable to visualize, it seems like it would also be valuable to symbolize localities by the date (or approximate time frame) of the observation.

·

Basic reporting

Overall, I found this to be a well-written paper that was mostly clear and straightforward. Sufficient background and context was presented and it was self-contained with mostly relevant results. The figures and tables were a bit confusing at first, but overall professionally prepared.

Experimental design

No comment

Validity of the findings

I did not find any major issues with the approach, analysis, or conclusions except for one. The conclusions and references to using the approach to assess losses in biodiversity are tenuous – you’ve not really demonstrated that either biodiversity is changing or that fireflies are a legitimate proxy for biodiversity. You can infer that some or all of the 26 known lampyrid species have declined over time, but we don’t actually know whether changes in abundance are caused by changes in one species or all of them. In particular, we don’t know what their individual abundances were prior to urbanization. If the authors wish to make this argument, they need to better explain the connection between loss of fireflies and loss of biodiversity.

Lines 260-261: You wrote: “The time of firefly sightings was significantly associated with the different Morelian generations X2= 22.553, df=10, p=0.012).” Wouldn’t this be an artifact of the different ages of participants? That is, a child less than 10 years old could not remember something from before they were born. If I am misunderstanding the statistical analysis or the statement, please clarify more explicitly what was tested and clarify the results sentence.

Lines 266-270: You wrote: “Moreover, in the oldest generation, the perspective of firefly loss was significantly higher than in the other generations X2= 21.201, df=8, p=0.006; Fig. 4b). A notable 20.8% of this generation reported a noticeable decline in firefly populations over time, compared to only 6.9% in the middle generation and 2.7% in the youngest generation (Fig. 4d).” This seems to me to be another artifact of the analysis – can you really compare a decline over time over different generations? The youngest generation could not have a perspective of long term loss, except if it was relayed to them, but that is another aspect of your analysis. I wonder if this should just be dropped, as it seems an inappropriate test/analysis.

Additional comments

Minor edits and clarifications
Line 112: You wrote: “Recently, Pérez-Hernández et al. (2023) found that urbanization in Morelia has affected…” – is the citation here 2023a or 2023b?

Lines 112-114: You wrote: “Recently, Pérez-Hernández et al. (2023) found that urbanization in Morelia has affected firefly populations in the region since, although there are a high number of firefly species (26) in and around the city, there are also several areas where these insects are locally extinct.” This sentence is awkwardly written. Try “Recently, Pérez-Hernández et al. (2023) found that urbanization in Morelia has affected firefly populations in the region. In particular, although there are a high number of firefly species (26) in and around the city, there are also several areas where these insects are known to be locally extinct.”

Lines 171-174: You wrote: “Ethical approval for this research was obtained from the Committee of Bioethics, Ethics in investigation, investigation, and Biosecurity, and by General Coordinator of the postgraduate programs of the Michoacana University of San Nicolás de Hidalgo Ethical to carry out this study.” This is a very confusing sentence. It is unclear whether the first is one committee or if there are two committees involved. Also, “investigation” is used twice. So, is it “the Committee of Bioethics, Ethics in Investigation and Biosecurity” or is it “the Committee of Bioethics, and Ethics in Investigation and Biosecurity” or something else entirely?

Lines 300-301: You wrote: “Furthermore, 61.8% (55 individuals; n=89) considered that the presence of fireflies in their locality is unfeasible.” What do you mean unfeasible? Do you mean it is not considered feasible that fireflies could live in those locations due to lack of habitat requirements or pollution or something else? Maybe edit as follows: “Furthermore, 61.8% (55 individuals; n=89) considered that it was not realistic or likely to have the presence of fireflies in their locality.”

Lines 347-348: You wrote: “…15 (12.9%) from the Naturalista (2016-2022) platform, and 15 (12.9%) from the entomological collections of the “Luciérnagas de Michoacán” project…” You refer to Naturalista multiple times, can you provide a URL on first use, in addition to the citation in the literature cited section? Also, out of curiosity, I searched GBIF.org and found records of Lampyridae in and around Morelia, which include both iNaturalist and museum records. I’m not sure of the redundancy to your data, but this could offer an additional source of data for you, long-term or for inclusion in this study.

Line 424: You use the term affectivity.– what do you mean by this? For non-social scientists, it might be helpful to define the term.

Lines 429-432: You wrote: “… sites and the gradual loss of fireflies were perceived mainly by individuals of more than 20 years in age (from the first and second generations). This is possibly because these individuals possess more information, having experienced past conditions and being aware of the changes that their environment has undergone, compared to individuals of the third generation – this is more because they’ve simply lived longer, right? Also, should be “changes that their environment have undergone”

Line 482: You wrote: “… cultural impact of the extinction of insects such as fireflies, x) documenting and reporting the…” – edit to add “and”: “…cultural impact of the extinction of insects such as fireflies, and x) documenting and reporting the…”

Lines 493-494: You use the term Baseline Change Syndrome – but previously, you used Shifting Baseline Syndrome. Be consistent.

Reviewer 3 ·

Basic reporting

The manuscript, "Oral History as a Citizen Science Tool to Understand Biodiversity Loss and Environmental Changes: A Case Study on Firefly Extinction in Morelia, Michoacán, Mexico," represents a valuable contribution to the conservation of fireflies in Michoacán. By emphasizing the role of citizen science and oral history, the study highlights how engaging local communities can drive meaningful change. Conservation biology often relies on policy changes and public participation to succeed, and this research underscores the importance of involving people in efforts to protect biodiversity.
However, the conclusions and methodology in this manuscript may be subject to bias, potentially leading to inaccurate results. For instance, shifts in people's perceptions could be influenced by changes in their habits or activities rather than an actual decline in the firefly population. While this issue is briefly mentioned at the end of the conclusion, I recommend expanding on it further. Specifically, a more thorough discussion of potential biases in data collection and the methods used to address them would strengthen the study’s credibility and reliability.
Additionally, the specific firefly species native to Michoacán were not identified or mentioned.

Experimental design

I recommend expanding the discussion on the experimental design to address and mitigate potential biases in data interpretation

Validity of the findings

The data interpretation may have certain biases; however, given the limited studies on fireflies in the region, engaging local communities in conservation efforts could play a pivotal role in influencing public policies and fostering the development of effective strategies to protect these species.

---

## Round 0.2 · Minor Revisions

Dear Dr. Pérez-Hernández,

after this new review round, the reviewer who read your text believes your manuscript may be accepted for publication in PeerJ after minor reviews. Please check the issues raised and proceed with the indicated suggestions.

Sincerely,

Daniel Silva

·

Basic reporting

The most recent revisions made by the authors has improved the clarity of the article.

I made a few comments about references that do not actually support the point being made (line 67).

Thank you to the authors for adding the raw questionnaire data in the original Spanish in the supplemental materials.

While the article is not framed with a single question and hypothesis, the area of inquiry and results are now framed more clearly and specifically than before.

Changes made to the figures generally made them significantly easier to read and interpret, particularly Figures 4, 5, and 6.

I continue to find the Figure 1A and 1B maps difficult to process/interpret. I suggest thinking about what the reader should take away/interpret from these maps and then change the design accordingly. Part of the challenge is that for the sighting report locations, the variables to symbolize are year of sighting and generation of respondent, while for verified records, the variables to symbolize are dataset (iNaturalist, entomological sampling) and year. I previously suggested using a color ramp for symbolizing year, but my current suggestion would be to separate the maps into additional panels that each contain fewer variables. This would allow the reader to more easily see the patterns that the authors would like to highlight (ie. older generations reporting more sightings from the peri-urban areas, younger generations reporting more sightings from the urban area, and verified persistence of fireflies in urban parks). This could look like a 3-panel map, with each panel corresponding to a generation, plus a map that only shows verified records symbolized by year and dataset.

I thought that Figure 3 was a very nice addition.

Experimental design

The authors have clarified various aspects of the methodology and provided additional helpful details.

There remain limitations to the types of inferences that can be drawn from the results because of the methodology, but the authors acknowledge and discuss these.

Validity of the findings

The authors have more carefully the inferences that can be made from the results, and also provided context for how the approach of using oral history methods could be applied in other areas for documenting potential firefly declines.

Additional comments

See various comments in the annotated PDF.

I commend the authors for all the improvements made to this article.

My main suggestion would be make the geographic and generational patterns in Figure 1 easier for the reader to see and absorb.

---

## Round 0.3 · accepted · Accept

Dear Dr. Pérez-hernandez,

After this new review round, I am pleased to accept your manuscript for publication in PeerJ! Congratulations!

Sincerely,

Daniel Silva

·

Basic reporting

No comment.

Experimental design

No comment.

Validity of the findings

No comment.

Additional comments

Thank you to the authors for addressing previous comments.